# Development of a Predictive Model of Occult Cancer After a Venous Thromboembolism Event Using Machine Learning: The CLOVER Study

**DOI:** 10.3390/medicina61010018

**Published:** 2024-12-27

**Authors:** Anabel Franco-Moreno, Elena Madroñal-Cerezo, Cristina Lucía de Ancos-Aracil, Ana Isabel Farfán-Sedano, Nuria Muñoz-Rivas, José Bascuñana Morejón-Girón, José Manuel Ruiz-Giardín, Federico Álvarez-Rodríguez, Jesús Prada-Alonso, Yvonne Gala-García, Miguel Ángel Casado-Suela, Ana Bustamante-Fermosel, Nuria Alfaro-Fernández, Juan Torres-Macho

**Affiliations:** 1Department of Internal Medicine, Hospital Universitario Infanta Leonor–Virgen de la Torre, 28031 Madrid, Spain; 2Venous Thromboembolism Unit, Hospital Universitario Infanta Leonor–Virgen de la Torre, Gran Via del Este Avenue, 80, 28031 Madrid, Spain; 3Department of Internal Medicine, Hospital Universitario de Fuenlabrada, 28942 Madrid, Spain; 4Venous Thromboembolism Unit, Hospital Universitario de Fuenlabrada, 28942 Madrid, Spain; 5Department of Internal Medicine, Clínica Universidad de Navarra-Hospital, 31008 Pamplona, Spain; 6Department of Internal Medicine, Hospital Universitario 12 de Octubre, 28041 Madrid, Spain; 7Department of Anatomical Pathology, Hospital Universitario Infanta Leonor–Virgen de la Torre, 28031 Madrid, Spain; 8Horus-ML, Alcalá Street 268, 28027 Madrid, Spain; 9Faculty of Medicine, Universidad Complutense de Madrid, 28040 Madrid, Spain

**Keywords:** early detection of cancer, machine learning, occult malignancy, predictive model, venous thromboembolism

## Abstract

*Background and Objectives*: Venous thromboembolism (VTE) can be the first manifestation of an underlying cancer. This study aimed to develop a predictive model to assess the risk of occult cancer between 30 days and 24 months after a venous thrombotic event using machine learning (ML). *Materials and Methods*: We designed a case–control study nested in a cohort of patients with VTE included in a prospective registry from two Spanish hospitals between 2005 and 2021. Both clinically and ML-driven feature selection were performed to identify predictors for occult cancer. XGBoost, LightGBM, and CatBoost algorithms were used to train different prediction models, which were subsequently validated in a hold-out dataset. *Results*: A total of 815 patients with VTE were included (51.5% male and median age of 59). During follow-up, 56 patients (6.9%) were diagnosed with cancer. One hundred and twenty-one variables were explored for the predictive analysis. CatBoost obtained better performance metrics among the ML models analyzed. The final CatBoost model included, among the top 15 variables to predict hidden malignancy, age, gender, systolic blood pressure, heart rate, weight, chronic lung disease, D-dimer, alanine aminotransferase, hemoglobin, serum creatinine, cholesterol, platelets, triglycerides, leukocyte count and previous VTE. The model had an ROC-AUC of 0.86 (95% CI, 0.83–0.87) in the test set. Sensitivity, specificity, and negative and positive predictive values were 62%, 94%, 93% and 75%, respectively. *Conclusions*: This is the first risk score developed for identifying patients with VTE who are at increased risk of occult cancer using ML tools, obtaining a remarkably high diagnostic accuracy. This study’s limitations include potential information bias from electronic health records and a small cancer sample size. In addition, variability in detection protocols and evolving clinical practices may affect model accuracy. Our score needs external validation.

## 1. Introduction

Prior studies have suggested an interrelation between cancer and venous thromboembolism (VTE) [1,2,3,4,5,6]. According to data from the Vienna Cancer and Thrombosis Study, the 2-year cumulative incidence of VTE is 8.7% in patients with cancer [6,7]. In comparison to the general population, a seven-fold increased risk of VTE has been reported in all age groups in patients with cancer [3,4,5,8,9]. Furthermore, prior studies have shown that the incidence of VTE depends on cancer type and is associated with a poor prognosis [3,4,5,6,10]. The hypercoagulable state in cancer patients is multifactorial in etiology. Cancer cells are recognized for producing procoagulant molecules, including tissue factor (TF), which can directly initiate coagulation pathways [11]. Furthermore, relative immobilization, pro-thrombotic side effects of some oncospecific therapies (e.g., hormonal therapy, anti-angiogenic treatments, and chemotherapy), surgical interventions, and central venous catheter placement also contribute to this process.

While the risk of thrombosis in cancer patients is well documented, the likelihood of occult malignancy in patients presenting with apparently unprovoked VTE remains less well understood. Several studies have estimated that occult cancer is identified in approximately 5% of unprovoked VTE cases [12,13]. In a large observational study from the Danish National Registry that included 26,653 patients with deep venous thrombosis (VTE) or pulmonary embolism (PE), the occurrence of cancer was 6.5% [14]. In this study, the risk of a cancer diagnosis was substantially elevated during the first six months after the thrombotic event. A systematic review reported a 12-month prevalence of occult malignancy to be 10% and 2.6% in patients with unprovoked and provoked VTE, respectively [15]. Therefore, VTE appears to be a significant harbinger of underlying malignancy, and the earlier diagnosis of these cancers could significantly improve patient survival.

Two scores to assess cancer risk in patients with VTE have been developed: the RIETE and SOME scores [16,17]. Jara-Palomares et al. developed the RIETE score, identifying six independent predictors of occult malignancy at the time of VTE diagnosis, with a follow-up period of 24 months [16]. Ihaddadene et al., in a post hoc analysis of the SOME trial [13], reported that age ≥ 60 years, history of provoked VTE, and current smoking status were associated with occult cancer detection (SOME score) [17]. Nevertheless, both scores’ predictive performance was poor in external validation studies [18,19,20,21,22]. On the other hand, randomized control trials that compared extensive versus limited screening for occult malignancy after an unprovoked VTE did not observe a positive impact on cancer-related and overall mortality [13,23,24,25]. Pooling data analysis showed that the neoplasms detected during the initial screening were higher with the performance of an exhaustive strategy compared to standard testing (odds ratio [OR], 2.04; 95% confidence interval [CI], 1.30–3.21; *p* < 0.001). However, no improvement was observed in overall mortality (OR, 0.90; 95% CI, 0.45–1.79; *p* = 0.67) or cancer-related mortality (OR, 0.96; 95% CI, 0.40–2.30; *p* = 0.89) [26].

The risk of missing a potentially early-stage cancer diagnosis carries significant implications for patients, as early cancer detection can improve treatment outcomes and enhance overall survival rates [27]. In VTE, clinicians initially focus on excluding thrombophilia and other VTE-related causes, such as immobilization, hospitalization, or surgery. They must then decide on the most suitable approach for screening occult malignancy. Non-targeted screening methods, including computed tomography (CT), positron emission tomography/CT (PET/CT), and endoscopies, have not demonstrated a survival benefit for VTE patients if they are diagnosed with cancer. Moreover, extensive screening methods can be costly and emotionally challenging for patients. Consequently, there is a pressing need for new research to develop effective screening strategies for this population.

Artificial intelligence (AI) has recently gained popularity in medicine. Machine learning (ML) is a branch of artificial intelligence that aims to build computer systems that automatically learn to solve a task from historical data. Evidence for ML-based prediction models in the VTE area has been accumulating over the past few years [28,29,30,31]. Implementation of these models might potentially be an improvement compared to existing models. This study aimed to develop an ML-based model for predicting occult cancer in patients with VTE.

## 2. Materials and Methods

We conducted a multicenter, analytic, observational study following the STROBE recommendations for observational studies [32]. The study was approved by the Ethics Committee of the Hospital Universitario Clínico San Carlos (code 23/535-E) and carried out in accordance with the Declaration of Helsinki and Good Clinical Practice guidelines.

### 2.1. Patients and Study Setting

Patients from Hospital Universitario Infanta Leonor and Hospital Universitario de Fuenlabrada, both in Madrid, Spain, with symptomatic acute unprovoked or provoked VTE (deep vein thrombosis [DVT] or pulmonary embolism [PE]), confirmed by objective tests (compression ultrasonography or contrast venography for DVT and helical CT scan, ventilation–perfusion lung scintigraphy, or angiography for PE) were included. The study period was between 1 March 2005 and 30 September 2021. At the time of VTE diagnosis, screening for hidden malignancies was conducted by the guidelines prevailing at that time, and no tumors were detected.

We performed a case–control study in a cohort of patients with VTE. Enrolled patients were located through the prospective electronic registry of the venous thromboembolism units of both centers. Data quality was regularly monitored. All patients or their relatives gave written or verbal consent to participate in the registry. Demographic, clinical, analytical, and radiological variables at the time of the thrombotic event were collected from electronic medical records using a standardized form.

Patients diagnosed with cancer beyond the first 30 days after experiencing a VTE were identified as cases, and those with no cancer detected during the first two years after experiencing a VTE were identified as the control group. The cancer diagnosis was confirmed with a tissue biopsy or cytology. Patients with localized non-melanoma skin tumor, localized prostate cancer (T1/T2N0M0), in situ/intramucosal colon carcinoma, and in situ adenocarcinoma of the uterine cervix as malignancies detected during follow-up were excluded. Evidence shows patients diagnosed with these local-stage cancers have a very low risk of VTE [33]. The low tumor burden of these neoplasms makes a thrombogenic potential unlikely and, thereby, including them could have biased our results.

We assessed the sites of cancer according to sex and age subgroups, time from VTE to cancer diagnosis, cancer stage at diagnosis, and mortality in cancer patients during the study period. Using AI, we built a predictive score aimed at identifying those patients at increased risk of occult cancer.

### 2.2. Baseline Variables

Patients enrolled had data collected around the time of the VTE diagnosis. Analyzed variables included, but were not limited to, age, gender, weight, presence of coexisting clinical conditions such as smoker, hypertension, diabetes, dyslipidemia, chronic heart or lung disease, and stroke, risk factors for VTE, including recent immobility, surgery, or estrogenic therapy, the extent of the DVT (proximal or distal thrombosis) and the PE (central or peripheral lung arterial affection), clinical signs on admission including heart rate, respiratory rate and systolic blood pressure, and laboratory results at baseline that included hemoglobin levels, platelet and leukocytes counts, serum creatinine levels, liver transaminase level, and D-dimer level, at baseline, and major bleeding. All patients were managed according to the current clinical practice.

### 2.3. Statistical Analysis

As part of the pipeline followed to build the ML models (Figure 1), we tested several preprocessing steps to improve the analytical potential of the dataset. Among these steps, we created synthetic cases of the minority class (occult cancer group) via the Synthetic Minority Oversampling Technique (SMOTE), filled non-informed values via Multivariate Imputation by Chained Equations (MICE), and removed redundant variables, i.e., variables with an extremely high correlation with other predictor variables, and irrelevant variables, i.e., variables with an extremely low correlation with the dependent variable.

For our purpose, three ML families of models were tested: (a) Extreme Gradient Boosting (XGBoost): Gradient boosting models build an ensemble of low-complexity models, usually shallow classification trees. XGBoost is an improvement on the above. (b) Light Gradient Boosting Machine (LightGBM): A variation of XGBoost models where trees grow leaf-wise instead of depth-wise like in XGBoost. Both families of ML provide similar predictive performance, but LightGBM is generally faster and more memory efficient. (c) CatBoost: Another variation of the XGBoost model. CatBoost is specially optimized to give optimal predictive performance when the dataset contains a large number of categorical, i.e., qualitative, values.

In order to train and validate these models, as well as to choose the model with the best predictive ability for the problem analyzed in this study, the dataset was divided into different subsets. (a) Training: This dataset was used to build the model by selecting the parameters or weights that minimized a given loss function for this particular set. (b) Validation: ML models have a number of possible configurations or hyperparameters. The validation set was used to select the combination of model and hyperparameters that minimized the model error over these data. In many cases, this validation set was not used explicitly and instead a technique called cross-validation (CV) was employed, where the training dataset was divided into k subsets or folds. The model was then trained using k-1 of these subsets, and the remaining one was used as the validation set. This process was repeated k times until all subsets had played the role of validation on one occasion, and the average of the errors from these k iterations was used as the final validation error. We employed a 3-fold cross-validation (k = 3). We chose here to use this cross-validation technique. (c) Test: This test set was used as if it was an external validation set. The combination of model and hyperparameters that achieved the lowest error over validation was evaluated again over the test set and the resulting metric was used as a measure of the expected error of the model built when used in a real clinical scenario. To address the possible imbalance in the dataset, we implemented several techniques to mitigate its impact. We applied oversampling techniques, mainly the Synthetic Minority Oversampling Technique (SMOTE), to create synthetic examples of the positive class over the training set and increase the number of positive rows during the model training. In addition, we used weighting techniques to increase the importance given by the model to the positive class and to increase the model’s focus on these cases. Of the patient cases, 75% were used for training and validation (including 75% of the patients with occult cancer) and 25% for testing. Of the patients employed for training and validation, 50% were used for training and 25% for validation at each CV iteration. This split into training, validation and test sets was implemented considering the temporal dimension, so the newest data were assigned to the test set.

To evaluate the discriminatory ability of the final model, we calculated the corresponding area under the receiver operating characteristic curve (ROC-AUC), sensitivity, specificity, positive predictive value (PPV), negative predictive value (NPV), and F1-score.

Mean and standard deviation summarize the quantitative variables, while we used the median for non-normally distributed data. Categorical variables were defined by frequency. Statistical analyses were performed using SPSS software, version 29.0 (SPSS, IBM Corp, Armonk, NY, USA).

## 3. Results

### 3.1. Characterization of the Study Patients

A total of 1123 patients were enrolled during the study period. Of these patients, 308 (27.4%) with cancer diagnosis at the time of VTE were excluded. Of the remaining 815 patients, 56 (6.9%) were diagnosed with malignancy beyond the first 30 days after VTE diagnosis (occult cancer group) and 759 were not diagnosed with cancer (control group) (Figure 2).

The median time between the VTE event and cancer diagnosis was 4.8 months. Baseline characteristics of patients are shown in Table 1. Half were men (51.5%), and their mean age was 59.3 ± 18 years. For 418 patients (51.3%), the VTE was considered unprovoked. Two in every three patients presented with PE (558/815; 68.4%) with or without DVT. Among 38 men with occult cancer, the most frequent sites were the prostate (19.6%), lung (10.7%), and gastrointestinal tract (10.7%). Among 18 women with occult cancer, the most common sites were the gastrointestinal tract (7.1%), breast (5.5%), hematologic sites (3.8%), and bladder (3.8%). Cancer was detected at an advanced stage (stages III and IV) in 22 patients (39.3%). A total of 19 (33.9%) patients died at the end of follow-up. All of them died as a result of cancer, except for one patient who died of abdominal sepsis.

### 3.2. Predictive Model Development

Of the 815 patients, 611 patients (75%) were used for training and validation (including 75% of the patients with occult cancer) and 204 (25%) for testing. From the 611 patients employed for training and validation, at each CV iteration, 407 patients (50%) were used for training and 204 (25%) for validation. A total of 121 variables were explored for the predictive analysis. CatBoost obtained better performance metrics among the ML models analyzed (Table 2). The final CatBoost model included, among the top 15 variables to predict hidden malignancy within the first two years following VTE diagnosis, patient’s demographic data (age and gender), vital signs on admission (systolic blood pressure and heart rate), comorbidities (weight and chronic lung disease), laboratory parameters on admission (D-dimer, alanine aminotransferase, hemoglobin, serum creatinine, cholesterol, platelets, triglycerides, and leukocyte count), and previous VTE. The importance obtained for each predictor is shown in Table 3. The hyperparameters of the winning CatBoost model are presented in Table 4. The final model had an ROC-AUC of 0.86 (95% CI, 0.83–0.87) in the test set (Figure 3). To further illustrate the model’s predictive ability, the confusion matrix for the test set is shown in Figure 4. This matrix demonstrates the percentage of correct and incorrect classifications made by the final model, with 85.3% true negatives, 8.3% true positives, 3.9% false negatives, and 2.5% false positives.

## 4. Discussion

In this study, we selected patients with a diagnosis of VTE in a real-world setting and developed a predictive model to assess the risk of occult cancer within 24 months in these patients. A total of 15 predictors of hidden malignancy were identified, including demographic data, comorbidities, and laboratory parameters.

In our study, 6.9% of patients with VTE and unknown cancer were diagnosed with malignancy between 30 days and 24 months after the thrombotic event. These results are aligned with previous publications [16,34]. In the RIETE score study, the proportion of patients with occult cancer was 7.6% [16]. Previously, Ihaddadene et al., in the SOME rule derivation study, found a lower proportion of patients with occult cancer (3.9%; 95% CI, 2.8–5.4); however, their mean age was lower than in our study (53 versus 59 years, respectively) [17].

In our series, most cancers were diagnosed within the first six months following the thrombotic event, consistent with previously published data [14,16,17,35]. The most common sites of cancer in men were the prostate, lung, and gastrointestinal tract. For women, occult cancers most frequently were located in the gastrointestinal tract, breast, hematologic sites, and bladder. These data agree with what has been previously reported [16,34]. This is crucial to decide the most suitable diagnostic approach for each patient.

The link between VTE and cancer is well documented. While VTE typically occurs in the advanced stages of cancer, the thrombotic event can also precede the onset of malignancy symptoms, potentially enabling an early diagnosis. From a theoretical point of view, early discovery of an occult neoplasm should improve prognosis, allowing prompt treatment initiation. Our model includes easy-to-obtain clinical and analytical variables readily available for VTE diagnosis.

Several tools have been developed to evaluate the risk of occult malignancy in patients with VTE. Jara-Palomares et al. proposed the RIETE score, which identifies six independent predictors of hidden cancer between 30 days and 24 months after a VTE event (provoked or unprovoked). These predictors include male sex, age over 70 years, chronic lung disease, anemia (hemoglobin levels < 13 g/dL for men and <12 g/dL for women), elevated platelet count (≥350,000 × 1000/mm^3^), prior VTE, and recent surgery [16]. Patients scoring ≤ 2 are classified as low risk, while those scoring ≥ 3 are categorized as high risk. In the original cohort, cancer was diagnosed during follow-up in 6.0% (95% CI, 5.1–6.6) of low-risk patients and 12.0% (95% CI, 10.4–13.5) of high-risk patients. Ihaddadene et al., in a post hoc analysis of the SOME trial, identified age ≥ 60 years, previous provoked VTE, and current smoking as factors associated with occult cancer detection within one year of an unprovoked VTE diagnosis [17]. Patients scoring ≤ 1 are considered low risk, while those scoring ≥ 2 are classified as high risk. These rules offer a promising approach for occult cancer screening in patients with VTE. Nevertheless, both scores’ predictive performance was poor in external validation studies [18,19,20,21,22], and no international clinical guidelines include recommendations for detecting occult cancer in VTE patients based on these scores. Some of the parameters included in the RIETE and SOME scores, such as age, hemoglobin, platelet count, chronic lung disease, and previous VTE, are also included in our model. Age is one of the main risk factors of cancer [36]. Although occult cancer can occur at all ages, the incidence is higher in the population in the age group 65 years and older [37]. The median age for diagnosis of the primary tumors common (lung, gastrointestinal, lymphoma, leukemia, and bladder) to both, males and females, is during peoples’ 70s. For prostate cancer, the median age is 79 years, while for female breast cancer, the median age is 71 years for each tumor [37]. These cancer statistics highlight an increasing incidence in older age groups. In patients with VTE, male sex has been associated with an increased risk of occult cancer. In a large cohort of patients with acute VTE from the Registro Informatizado Enfermedad TromboEmbolica registry (RIETE), 7.6% (444/5864) of patients had occult cancer. Males had an increased risk of lung, prostate, and colorectal cancer. Among women, the risk of cancer was lower, and the most common sites included the colon, breast, uterus, pancreas, and hematologic sites [38]. Blood-based biomarkers, including hemoglobin, platelet count, and leukocyte count, can provide clues to the presence of occult cancer. On the other hand, interestingly, our model yielded some predictors that are related factors for malignancy, such as nutritional parameters, and they had not previously been associated with occult cancer. Tumor cells in an active proliferative or metastatic state depend on lipid metabolism. A recent study has identified several genes involved in lipid metabolism in cancer patients [39]. This lipid metabolic reprogramming plays an essential role in oncogenesis [40], highlighting lipids’ potential role in detecting occult cancer in VTE patients. Moreover, in the test cohort, the ML-based model demonstrated superior performance compared to the RIETE score, achieving a higher ROC-AUC (0.86 versus 0.64, respectively).

In our model, smoking was not found among the 15 key predictors for hidden malignancy. Chronic lung disease is a surrogate for smoking, and smoking is associated with an increased risk of cancer. Hence, the higher risk of occult cancer in patients with chronic lung disease might likely be related to tobacco consumption.

The CLOVER score does not set explicit thresholds for numerical variables. ML models naturally handle continuous variables by assigning weights to them, without the need for arbitrary thresholds. Setting a threshold on the continuous variables would convert them into categorical or binary variables, leading to a loss of information, which could be critical for model accuracy. ML models capture and utilize the full range of information in the data by directly analyzing continuous values, leading to better performance.

It is essential to address the implications of the model’s false negatives (3.9%), where the model fails to predict an occult malignancy. Given the critical nature of these cases, all patients classified as low-risk (negative predictions) should be followed closely for up to two years to confirm that no malignancy develops. This follow-up period aligns with the clinical standard for patients with VTE at risk of hidden malignancies, ensuring that any potential tumors missed by the model are detected during routine assessments.

Variations in protocols across different institutions might influence model predictions. If significant differences are found in a specific institution, a new specialized version of the model, fine-tuned to focus on the patients of that institution, could be trained. This way, we could have a “generic” model and then specialized models for different institutions, which will be trained to take into account the unique characteristics of the cohort of patients of each particular institution.

This study has some limitations. First, as the results rely on the accuracy of the electronic health records reflected by physicians in clinical practice, they may be susceptible to information bias. In this regard, not all the study variables of interest were accessible in all patient records, leading to missing patient data. Second, the sample size of patients who developed cancer may be insufficient; however, the proportion of patients presenting later with occult cancer and the most common sites of cancer agree with those reported in previous studies. Third, sometimes, ML models are not explanatory; nevertheless, the variables incorporated into our model hold clinical significance and appear pertinent to this clinical context. Fourth, the strategies to detect hidden malignancies in patients with VTE changed over the study period. Therefore, patients did not undergo a single protocol. ML predictive models inherently rely on historical data. We recognize that changes in research parameters (such as the incorporation of genetic factors), clinical practices, and external factors influencing the population’s demographic characteristics and the most frequent type of tumors could affect the ML patterns and impact the model’s predictive accuracy. To mitigate this risk, performing periodic retraining using updated datasets to ensure adaptability to evolving clinical and demographic trends would be appropriate. Finally, despite using derivation and validation cohorts, external validation is necessary to confirm the model’s accuracy. Validating the model in an independent cohort is essential to ensure its reliability and applicability. Future prospective, multicenter cohort studies are needed to guarantee its robustness in the real world.

## 5. Conclusions

To the best of our knowledge, this is the first ML predictive model designed to predict occult cancer in patients with VTE between 30 days and 24 months after the thrombosis event (develop and test validation). The score can be used easily. External validation in the near future would be critical to extrapolate the use of the proposed model and to identify patients who may benefit from extensive screening for malignancy.

## Figures and Tables

**Figure 1 medicina-61-00018-f001:**
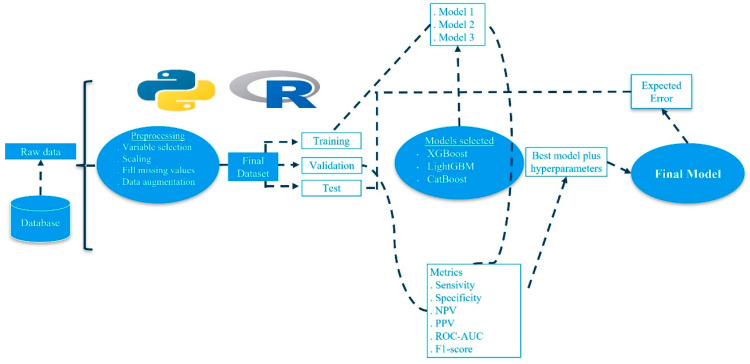
Machine learning pipeline. Abbreviations: PPV, positive predictive value; NPV, negative predictive value; ROC-AUC, area under the receiver operating characteristic curve.

**Figure 2 medicina-61-00018-f002:**
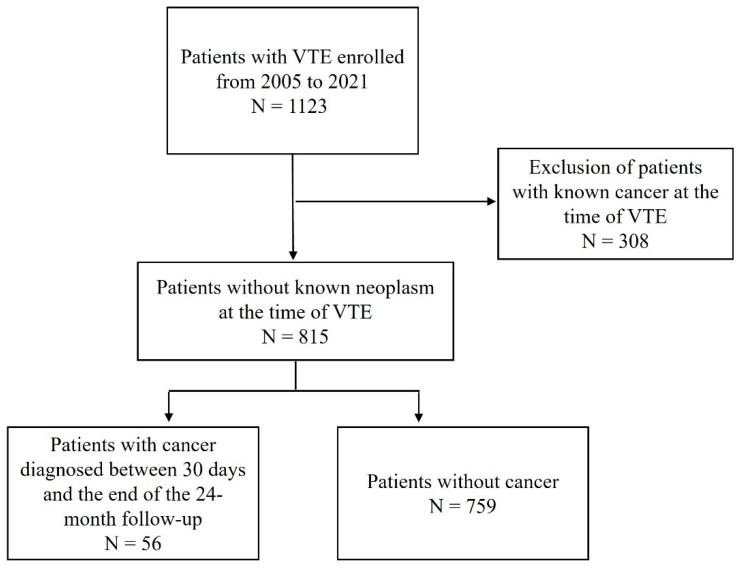
Flowchart of patients. Abbreviations: VTE, venous thromboembolism.

**Figure 3 medicina-61-00018-f003:**
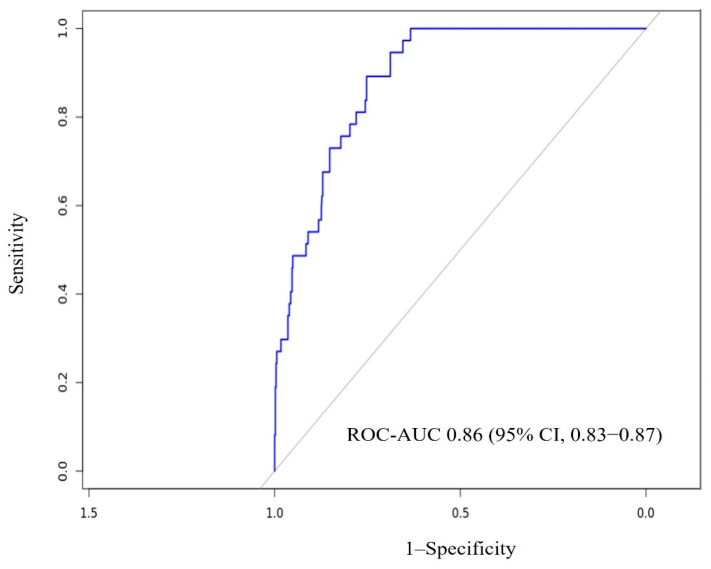
ROC-AUC of the final model according to the test set.

**Figure 4 medicina-61-00018-f004:**
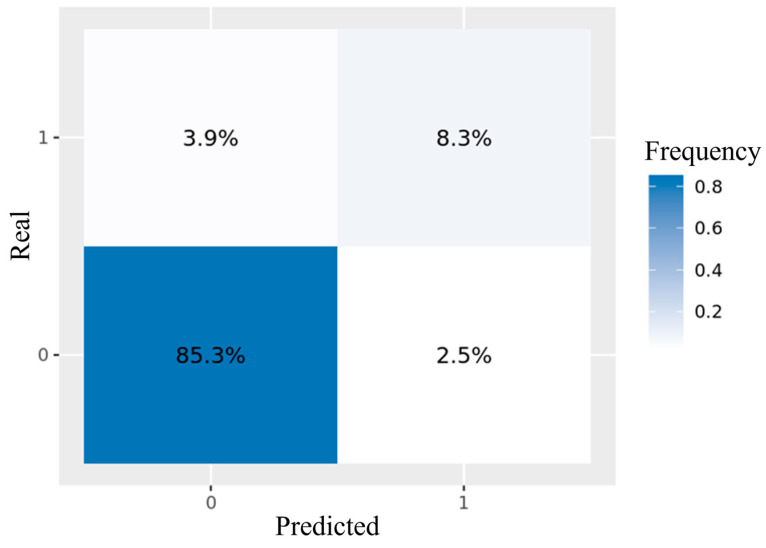
Confusion matrix of the final CatBoost model in the test set.

**Table 1 medicina-61-00018-t001:** Demographics, comorbidities, clinical characteristics, and laboratory findings of patients at the time of venous thromboembolism diagnosis.

	Total N = 815
**Demographic Characteristic**	
Age—years (median [SD])	59.3 (±18)
Male—n (%)	420 (51.5)
**Comorbidities—n** (**%**)	
Obesity (BMI > 30 kg/m^2^)	314 (38.5)
Hypertension	332 (40.7)
Diabetes mellitus	106 (13.0)
Dyslipidemia	101 (12.4)
Smoking	177 (21.7)
Chronic lung disease	77 (9.4)
Chronic heart disease	23 (2.8)
Autoimmune disease	2 (0.2)
**Initial VTE presentation—n** (**%**)	
PE with or without DVT	558 (68.4)
DVT	257 (31.6)
**Unprovoked VTE—n** (**%**)	418 (51.3)
**Vital signs on admission—median** (**SD**)	
Heart rate—bpm	87 (±18)
Respiratory rate—bpm	12 (±5)
Systolic blood pressure—mmHg	133 (±22)
**Laboratory parameters on admission—median** (**IQR**)	
Hemoglobin—g/dL	13.8 (12.7–15.1)
Leukocytes—cells/mL	9546 (7100–11,300)
Platelets × 1000/mm^3^	240 (180–284)
Creatinine—mg/dL	0.96 (0.75–1.10)
D-dimer—ng/mL	5655 (1127–5866)
Fibrinogen—mg/dL	569 (500–594)
Troponin—ng/L	1.21 (0.14–0.60)
NT-proBNP—pg/mL	2964 (148–3861)

Abbreviations: BMI, Body Mass Index; DVT, deep vein thrombosis; IQR, interquartile range; NT-proBNP, N-terminal pro-B-type natriuretic peptide; PE, pulmonary embolism; SD, standard deviation; VTE, venous thromboembolism.

**Table 2 medicina-61-00018-t002:** Model performance metrics in the test set.

ML Model	Metric					
	Sensitivity	Specificity	NPV	PPV	ROC-AUC	F1-Score
CatBoost	0.62	0.94	0.93	0.75	0.86	0.68
XGBoost	0.48	0.89	0.81	0.66	0.76	0.57
LightGBM	0.58	0.92	0.88	0.70	0.82	0.64

Abbreviations: ML, machine learning; NPV, negative predictive value; PPV, positive predictive value; ROC-AUC, area under the receiver operating characteristic curve.

**Table 3 medicina-61-00018-t003:** Fifteen most important variables for the final predictive model for occult cancer in patients with venous thromboembolism.

Variable	Relative Importance
Age—years	1
D-dimer—ng/mL	0.71
Systolic blood pressure—mmHg	0.66
Alanine aminotransferase (ALT)—U/L	0.61
Hemoglobin—g/dL	0.59
Creatinine—mg/dL	0.51
Total cholesterol—mg/dL	0.50
Platelets × 1000/mm^3^	0.45
Triglycerides—mg/dL	0.33
Leukocytes—cells/mL	0.25
Weight—kilograms	0.18
Chronic lung disease	0.16
Heart rate	0.14
Gender	0.14
Previous venous thromboembolism	0.14

**Table 4 medicina-61-00018-t004:** Hyperparameters of the final CatBoost model.

Hyperparameter	Value
learning_rate	0.1
rsm	0.1
depth	6
l2_leaf_reg	3
border_count	8
min_data_in_leaf	100
bootstrap_type	Bayesian
bagging_temperature	10
subsample	0.66
sampling_frequency	PerTreeLevel
sampling_unit	Object
grow_policy	Depth-wise
boosting_type	Ordered
model_shrink_rate	0.5
model_shrink_mode	Constant
best_iteration	873

## Data Availability

The data supporting this study’s findings are available from the corresponding author upon reasonable request. Restrictions apply to the availability of these data, which were used under license for this study.

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
