# Peer review of "Development of a Predictive Model of Occult Cancer After a Venous Thromboembolism Event Using Machine Learning: The CLOVER Study"

_medicina, 2024, doi:10.3390/medicina61010018_

Round 1
Reviewer 1 Report
Comments and Suggestions for Authors
In the article, an ML-based predictive model is proposed to evaluate the latent cancer risk within 30 days and 24 months after a venous thrombotic event. I have a few suggestions,
1. 15 features from the demographic characteristics, comorbidities, clinical characteristics, and laboratory findings of the patients were used for the model. For example, a high hemoglobin value may be related to cancer. However, how are demographic characteristics related to cancer? Please elaborate.
2. What is the k value selected? Are the results obtained average results? (table 2)
3. Add the confusion matrix to the article. Especially comment on FN and FP. It is dangerous to say that a sick individual is not sick.
4. Add the loss and accuracy changes in the training process of your model to the article. These changes indicate that the training is complete. If you have not recorded these changes. Add the hyperparameters for your models used in the form of a table to the article.
5. There is a data imbalance in your data set. One of the reasons why the Model Sen, PPV, and F1 parameters are not high is data imbalance. Please discuss this critical problem.
Author Response
Response to reviewer' comments
15 features from the demographic characteristics, comorbidities, clinical characteristics, and laboratory findings of the patients were used for the model. For example, a high hemoglobin value may be related to cancer. However, how are demographic characteristics related to cancer? Please elaborate.
We have added in discussion section the following paragraph: Age is one of the main risk factors of cancer [37]. Although occult cancer can occur at all ages, the incidence is higher in the population in the age group 65 years and older. [38]. The median age at diagnosis for the primary tumors common to both males and females (lung, gastrointestinal, lymphoma, leukaemia, and bladder) occurs during the 1970s. For prostate cancer, the median age is 79 years, while for female breast cancer, the median age is 71 years for each tumor. These cancer statistics highlight an increasing incidence in older age groups [38]. In patients with VTE, male sex has been associated with an increased risk of occult cancer. In a large cohort of patients with acute VTE from the Registro Informatizado Enfermedad TromboEmbolica registry (RIETE), 7.6% (444/5864) of patients had occult cancer. Males had an increased risk of lung, prostate and colorectal cancer. Among women, the risk of cancer was lower, and the most common sites included the colon, breast, uterus, pancreas, and hematologic.
What is the k value selected? Are the results obtained average results? (table 2)
The k value selected for CV is 3 (as indicated in methods, page 8-9, 50% of the train_val set is used for train, and 25% for validation in each CV iteration). However, the results obtained in Table 2 are not the CV metrics, but the metrics over the test set, a split independent of train and validation used for the final evaluation of the model, as recommended by machine learning good practices. We have added in the methods section this specific data: We employed a 3-fold cross-validation (k =3).
Add the confusion matrix to the article. Especially comments on FN and FP. It is dangerous to say that a sick individual is not sick.
In results, we have added the confusion matrix in the results section.
To address this issue, we have included it in the discussion section: It is essential to address the implications of the model's false negatives (3.9%), where the model fails to predict an occult malignancy. Given the critical nature of these cases, all patients classified as low-risk (negative predictions) should be followed closely for up to two years to confirm that no malignancy develops. This follow-up period aligns with the clinical standard for patients with venous thromboembolism (VTE) at risk of hidden malignancies, ensuring that any potential tumors missed by the model are detected during routine assessments.
Add the loss and accuracy changes in the training process of your model to the article. These changes indicate that the training is complete. If you have not recorded these changes. Add the hyperparameters for your models used in the form of a table to the article.
We did not record that information. In fact, we used early stopping, a training technique that allows to detect when your validation error starts increasing and automatically stops the learning process, as continuing the training of the model will be incorrect event if the loss and accuracy over training keep improving. Therefore, we add the hyperparameters of our winner model:
learning_rate 0.1
rsm 0.1
depth 6
l2_leaf_reg 3
border_count 8
min_data_in_leaf 100
bootstrap_type Bayesian
bagging_temperature 10
subsample 0.66
sampling_frequency PerTreeLevel
sampling_unit Object
grow_policy Depthwise
boosting_type Ordered
model_shrink_rate 0.5
model_shrink_mode Constant
best_iteration 873
There is a data imbalance in your data set. One of the reasons why the Model Sen, PPV, and F1 parameters are not high is data imbalance. Please discuss this critical problem.
It is clear that there exists an imbalance problem in the data. We have applied the following techniques to try to lessen the impact of this issue:
Oversampling: We have applied oversampling techniques, in particular Synthetic Minority Over-sampling Technique (SMOTE) , to create synthetic examples of the positive class over the train set, to increase the number of positive rows during the training of the model.
Weighting: We have used weighting techniques to increase the importance given by the model to the positive class, to try to increase the focus of the model on these cases.
We have added in the methods section: To address the possible imbalance in the dataset, we implemented several techniques to mitigate its impact. We applied oversampling techniques, mainly the Synthetic Minority Over-sampling Technique (SMOTE), to create synthetic examples of the positive class over the train set and increase the number of positive rows during the model training. In addition, we used weighting techniques to increase the importance given by the model to the positive class and to increase the model's focus on these cases.
Reviewer 2 Report
Comments and Suggestions for Authors
This is a well written manuscript, which is based on the investigating the role of Venous thromboembolism in appearance of an underlying cancer. This study focused on developing a predictive model to assess the risk of occult cancer after a venous thrombotic event using 3 different Machine Learning Models.
The manuscript is of interest to internal medicine and clinical researchers in aspects of investigating increased risk for occult cancer and identifying the patients with VTE by using Machine Learning tools, which has a significantly high diagnostic accuracy.
In my opinion, this study approach provides much useful information about the extrapolate use of the proposed model and to identify patients who benefited by extensive screening for malignancy.
I would recommend this study to be accepted after a few minor corrections and provide brief explanations:
1. How the variations in treatment protocols across different institutions can influence model predictions?
2. Predictive models are based on historical data, which assumes that past patterns and relationships will continue in the future. However, research study dynamics, research parameters, and external factors can change over time, which significantly influence the predictive outcome and makes the predictions less accurate. So, how do these factors influence this study?
3. The accuracy of predictive analytics models is limited by the completeness; accuracy of the data being used and difference in the performance metrics. So, how do authors validate the high diagnostic accuracy in this study?
4. Authors should focus more on the significance and usefulness of this study.
Thanks
Author Response
How the variations in treatment protocols across different institutions can influence model predictions?
This is a reality. We have discussed it in the limitations paragraph: Variations in protocols across different institutions might influence model predictions. If significant differences are found in a specific institution, a new specialized version of the model, fine-tuned to focus on the patients of that institution, could be trained. This way, we could have a "generic" model and then specialized models for different institutions, which will be trained to take into account the unique characteristics of the cohort of patients of each particular institution.
Predictive models are based on historical data, which assumes that past patterns and relationships will continue in the future. However, research study dynamics, research parameters, and external factors can change over time, which significantly influences the predictive outcome and makes the predictions less accurate. So, how do these factors influence this study?
This is a reality. We have discussed it in the limitations paragraph : ML-predictive models inherently rely on historical data. We recognize that changes in research parameters (such as the incorporation of genetic factors), clinical practices, and external factors influencing the population's demographic characteristics and the most frequent type of tumors could affect the ML patterns and impact the model's predictive accuracy. To mitigate this risk, performing periodic retraining using updated datasets to ensure adaptability to evolving clinical and demographic trends would be appropriate.
The accuracy of predictive analytics models is limited by the completeness, accuracy of the data being used and difference in the performance metrics. So, how do authors validate the high diagnostic accuracy in this study?
The models have been evaluated over a validation set and over a test set, playing the role of external validation. Metrics over the test set show a high diagnostic performance, and individual metrics, like sensitivity, could be further enhanced by worsening others, like specificity, if deemed necessary. In addition, we are already working in increasing the volume of the dataset employed, which is expected to increase the predictive performance of the model.
In addition, we are also planning on conducting a prospective validation of the tool, using it in the background a real clinical scenario in several institutions, so we can measure its predictive performance over future real patients without risking affecting the physicians’ decisions regarding patient treatment.
Authors should focus more on the significance and usefulness of this study
We have focused on the significance and usefulness of this study in the introduction section: The risk of missing a potentially early-stage diagnosis of malignancy has real implications for patients. Early detection of cancer will improve the success of patient treatment and enhance overall survival [40]. In the clinical scenario of VTE, clinicians act to rule out thrombophilia and other causes related to VTE, such as immobilization or surgery. They are then left in the position to determine appropriate screening for an occult malignancy. The non-targeted screening using computed tomography (CT), positron emission tomography/CT (PET/CT) and endoscopies does not provide patients with VTE any survival advantage even if they are diagnosed with cancer. An exhaustive screening can also be costly and psychologically draining. Therefore, new lines of research are needed to develop appropriate strategies for cancer screening in this patient population.
Reviewer 3 Report
Comments and Suggestions for Authors
The study represents a valuable contribution to the field of predictive oncology by exploiting ML to improve occult cancer risk stratification in VTE patients. Although the results are promising, the model requires further validation and refinement to ensure clinical applicability. Addressing the limitations would significantly improve the impact of the study and open excellent avenues for future research
Recommendations for improvement:
- Abstract: Include a brief mention of limitations to present a balanced overview.
- Introduction: Elaborate on the clinical significance of early detection of occult cancer in patients with VTE.
- Methods: Justify the exclusions of some cancer types and provide more details on the impact of imputation of missing data.
- Results: Discuss the clinical implications of low model sensitivity and trade-offs with specificity.
- Discussion: Explore limitations in more detail, include a discussion of real-world implementation challenges.
- Validation: Prioritise external validation and specify how this might be achieved in future studies.
Author Response
Abstract: Include a brief mention of limitations to present a balanced overview.
We have included the limitations in the abstract: This study's limitations include potential information bias from electronic health records and a small cancer sample size. In addition, variability in detection protocols and evolving clinical practices may affect model accuracy
Introduction: Elaborate on the clinical significance of early detection of occult cancer in patients with VTE
We have focused on the clinical significance of early detection of occult cancer in VTE patients in the introduction section: The risk of missing a potentially early-stage diagnosis of malignancy has real implications for patients. Early detection of cancer will improve the success of patient treatment and enhance overall survival [40]. In the clinical scenario of VTE, clinicians act to rule out thrombophilia and other causes related to VTE, such as immobilization or surgery. They are then left in the position to determine appropriate screening for occult malignancy. The non-targeted screening using CT, PET/CT and endoscopies does not provide patients with VTE any survival advantage even if they are diagnosed with cancer. An exhaustive screening can also be costly and psychologically draining. Therefore, new lines of research are needed to develop appropriate strategies for cancer screening in this patient population.
Methods: Justify the exclusions of some cancer types and provide more details on the impact of imputation of missing data.
We have included this information in the methods section: Evidence shows patients diagnosed with this local-stage cancers have a very low risk of VTE. The low tumor burden of these neoplasms makes a thrombogenic potential unlikely and, thereby, including them could have biased our results.
Imputation of missing data showed to be critical to improve predictive performance of the model. Using Multivariate Imputation by Chained Equations (MICE) improved the evaluation metrics over test around 10%.
Results: Discuss the clinical implications of low model sensitivity and trade-offs with specificity.
Aa for the tradeoff between sensitivity and specificity, it is important to remark that it is possible to improve one metric, at the cost of worsening another, by using different decision thresholds to decide what is ‘positive’ and what is ‘negative’, since the model outputs a score or probability of having the disease, not a direct 0/1 label. In our case, we have decided for this paper to choose as best model the one that maximized the F1-score, in order to select a kind of “agnostic” model in terms of this tradeoff, but for real implementations of the solution it may be desirable to put more focus on sensitivity.
To address this issue, we have included it in the discussion section on how to ensure the negative values properly: It is essential to address the implications of the model's false negatives (3.9%), where the model fails to predict an occult malignancy. Given the critical nature of these cases, all patients classified as low-risk (negative predictions) should be followed closely for up to two years to confirm that no malignancy develops. This follow-up period aligns with the clinical standard for patients with venous thromboembolism (VTE) at risk of hidden malignancies, ensuring that any potential tumors missed by the model are detected during routine assessments.
Discussion: Explore limitations in more detail, include a discussion of real-world implementation challenges
To address this issue, we have included it in the discussion section: ML-predictive models inherently rely on historical data. We recognize that changes in research parameters (such as the incorporation of genetic factors), clinical practices, and external factors influencing the population's demographic characteristics and the most frequent type of tumors could affect the ML patterns and impact the model's predictive accuracy. To mitigate this risk, performing periodic retraining using updated datasets to ensure adaptability to evolving clinical and demographic trends would be appropriate.
Validation: Prioritize external validation and specify how this might be achieved in future studies.
To address this issue, we have included it in the discussion section: Validating the model in an independent cohort is essential to ensure its reliability and applicability. Future prospective, multicenter cohort studies are needed to guarantee its robustness in the real world.